# Functional culture and in vitro genetic and small-molecule manipulation of adult mouse cardiomyocytes

Neal I. Callaghan[1,2,12], Shin-Haw Lee[2,3,12], Sina Hadipour-Lakmehsari[2,3,12], Xavier A. Lee[2,3], M. Ahsan Siraj[4,5], Amine Driouchi[1,6,7,8,11], Christopher M. Yip[1,6,7,8], Mansoor Husain[3,4,5,9], Craig A. Simmons[1,2,10] & Anthony O. Gramolini[2,3,4,9 ✉]

Primary adult cardiomyocyte (aCM) represent the mature form of myocytes found in the adult heart. However, culture of aCMs in particular is challenged by poor survival and loss of phenotype, rendering extended in vitro experiments unfeasible. Here, we establish murine aCM culture methods that enhance survival and maintain sarcomeric structure and $Ca^{2+}$ cycling to enable physiologically relevant contractile force measurements. We also demonstrate genetic and small-molecule manipulations that probe mechanisms underlying myocyte functional performance. Together, these refinements to aCM culture present a toolbox with which to advance our understanding of myocardial physiology.

---

[1] Institute of Biomaterials and Biomedical Engineering, Faculty of Applied Science and Engineering, University of Toronto, Toronto, ON, Canada. [2] Translational Biology and Engineering Program, Ted Rogers Centre for Heart Research, Toronto, ON, Canada. [3] Department of Physiology, Faculty of Medicine, University of Toronto, Toronto, ON, Canada. [4] Toronto General Hospital Research Institute, University Health Network, Toronto, Ontario, Canada. [5] Ted Rogers Centre for Heart Research, University Health Network, Toronto, Ontario, Canada. [6] Department of Biochemistry, University of Toronto, Toronto, ON, Canada. [7] Department of Chemical Engineering & Applied Chemistry, Toronto, ON, Canada. [8] Donnelly Centre, University of Toronto, Toronto, ON, Canada. [9] Heart and Stroke Richard Lewar Centre of Excellence, University of Toronto, Toronto, Ontario, Canada. [10] Department of Mechanical and Industrial Engineering, Faculty of Applied Science and Engineering, University of Toronto, Toronto, ON, Canada. [11] Present address: Institute for Biophysical Dynamics, and Department of Biochemistry and Molecular Biology, University of Chicago, Chicago, Illinois, USA. [12] These authors contributed equally: Neal I. Callaghan, Shin-Haw Lee, Sina Hadipour-Lakmehsari. ✉ email: anthony.gramolini@utoronto.ca

In vitro primary cell culture is a valuable tool to complement in vivo physiological investigation of many tissues. In vitro studies of myocardium are limited by the challenges of adult cardiomyocyte (aCM) culture. Although neonatal and pluripotent stem cell-derived cardiomyocytes have high survival rates in culture, they do not fully replicate the adult phenotype in terms of morphology, mature protein isoform expression, action potential component currents, $Ca^{2+}$ transient dynamics, contractile force production, or metabolic substrate preference[1,2]. Therefore, these immature cells fall short of replicating the physiology of mature CMs required for detailed mechanistic study[1]. While primary aCM isolation remains a crucial tool for acute studies[2], primary aCMs cultured in single-cell format detach and rapidly lose physiological function[2–4], with surviving cells typically assuming a 'fetal' phenotype after 2–3 weeks in culture[2]. Finally, aCM contractility has been primarily measured using shortening velocity of cells in suspension or isometric force production of intact or permeabilized cells glued to a force transducer. Neither of these methods represent a physiological setting. aCMs respond in real time to their physical environment[5,6], and commonly used contractile metrics of shortening velocity of aCMs in suspension are not ideal because they do not provide physiological resistance nor estimate contractile force directly. Here, we demonstrate a culture protocol that maintains primary murine aCM function, enables physiological contractile force measurement, and is amenable to genetic and small-molecule treatments with phenotype-altering effects.

Typically, primary aCMs are cultured on a purified matrix and supplemented with specific growth factors added to the media. Since aCM function depends on various integrin–ECM protein interactions to allow for precise regulation of downstream signaling pathways[7,8], we first explored altering the matrix constituents to promote aCMs. Basement membrane preparations including Geltrex provide a complex ECM protein mixture, including collagens, laminins, fibronectins, and entactins among others[9], recapitulating the diversity of composition of intact myocardial ECM[10]. Secondly, the strength of aCM contractions result in substantial stresses at the points of cell-substrate contact. Fully contracting aCMs rapidly detach from the substrate, necessitating the inhibition of contractility in aCM culture. Butanedione monoxime (BDM) was initially adopted as a myosin ATPase inhibitor, however, BDM also impairs $Ca^{2+}$ cycling through L-type channels[11,12], modulates cardiac ryanodine receptor (RyR) flux in a $Ca^{2+}$-dependent manner[13], and inhibits the oxidative metabolism upon which aCMs are reliant[14], all of which are deleterious to maintaining cellular myocyte homeostasis[15]. A non-BDM protocol to inhibit contractility, using a specific inhibitor of myosin ATPase, blebbistatin, has shown enhanced aCM longevity and function in culture[3]. In this study, we demonstrate the combinatorial benefit of both Geltrex and blebbistatin in aCM culture.

## Results

**Enhanced cell survival and function.** Previous methods attempting prolonged aCM survival have shown significant deterioration in culture with complete loss of initial sarcomere structure by day 8 (d8)[2], followed by regression to a rounded neonatal morphology. The functional hallmarks of the aCM, including $Ca^{2+}$ handling and contractility, are rapidly abrogated and can only effectively be assessed on the day of isolation; this loss of function precludes the use of experimental treatments, except for short-term, specific activators and inhibitors. As a result, the aCM model has failed to gain traction despite its unique ability to recapitulate characteristics of the native myocardium. We hypothesized that altering the culture conditions

would profoundly extend the utility of isolated myocytes, and we focused on enhancing the matrix with Geltrex together with inhibiting cellular contraction using blebbistatin. CMs isolated from adult mice and cultured using our modified protocol combining Geltrex and blebbistatin demonstrated high adhesion, survivability, and continued functionality in sustained culture (Fig. 1). aCMs retained α-actinin-positive sarcomeres (visualized by α-actinin) and an ordered sarcoplasmic reticulum (SR; visualized by confocal imaging for SERCA2, RyR2, DHPR α-subunit, and STIM1) with both longitudinal and cisternae components observed up to 7 days post isolation (Fig. 1a). DHPR-positive t-tubules exhibited a mild but progressive degree of altered expression patterns, starting on d1 after isolation, with the presence of discrete puncta observed within the myocyte. Next, we examined Akt total and phosphorylation levels, since Akt-dependent signaling is centrally involved in the regulation of cell differentiation, proliferation, and growth. Immunoblots were performed in isolated cells either cultured on standard laminin matrix, or with Geltrex; the latter resulted in a small decrease in Akt phosphorylation compared to a standard laminin coating ($32 \pm 11\%$ reduction; $p = 0.02$, Fig. 1b; full blots in Supplementary Fig. 1). To profile the cells, we carefully examined the morphology of the cardiomyocytes in culture and performed cell counts for the presence of rod-shaped cells, or the cells showing a rounded morphology, representing poor quality myocytes. Cell counts indicated that the Geltrex and blebbistatin treated cultures remained largely viable and showed a minor decline from their initial yield of c.a. 90% rod-shaped cells (Fig. 1c). In fact, surviving aCMs exhibited morphology similar to their initial state immediately post-isolation for up to 1 week in culture and did not show substantial alteration in size, shape, or general morphology, suggesting that this method allows for the isolation of stable and healthy adult cardiomyocytes.

We next compared all combinatorial effects of culture conditions (Geltrex or laminin, with blebbistatin or BDM) and determined which conditions resulted in the greatest cell survival with the maintenance of cellular morphology for the isolated aCMs. Myocyte isolations were performed using laminin as the conventional approach, with blebbistatin or BDM, and Geltrex with blebbistatin or BDM, with attached cell counts performed daily over 3 fields of view per replicate. In this detailed comparison, we showed that laminin-based preparations showed cellular preservation comparable to the Geltrex and blebbistatin combination during the 7 days; with a general decline in rod-shaped cells from c.a. 75% down to c.a. 20–25% of initial cells. Similar outcomes were seen whether BDM or blebbistatin were used. Interestingly, our results clearly show the combinatorial benefit of Geltrex and blebbistatin in culture. The use of Geltrex with BDM resulted in fewer than 20% of cells retaining a rod-shaped morphology at d7, with corresponding detection of rounded cells approaching 50% of the initial population (Fig. 2a, left panel). Consistent with the general morphological observations, confocal analysis of myocytes under these conditions showed pronounced degradation of sarcoplasmic staining expression patterns of DHPR in the laminin plus BDM or blebbistatin conditions, as well as in the Geltrex and BDM cells (Fig. 2b).

Spontaneous $Ca^{2+}$ transient properties were analyzed using Fluo-4 AM fluorescent microscopy (Fig. 3a, b) for up to 1 week in the Geltrex and blebbistatin prepared cells. The time to $Ca^{2+}$ transient peak and the time to 50% decay is shown in Fig. 3a and shows a stable time to peak of $162 \pm 6$ ms and $241 \pm 4$ ms at d0 through to $155 \pm 12$ ms and $266 \pm 19$ ms at d7, respectively. The peak amplitude of the $Ca^{2+}$ transients were seen to sharply decrease after d0 ($F/F_0 = 1.192 \pm 0.012$) at d1 ($F/F_0 = 1.09 \pm 0.007$) but then remained stable through to d7 (Fig. 3b). The general kinetics of the $Ca^{2+}$ transients tended to decrease after d1,

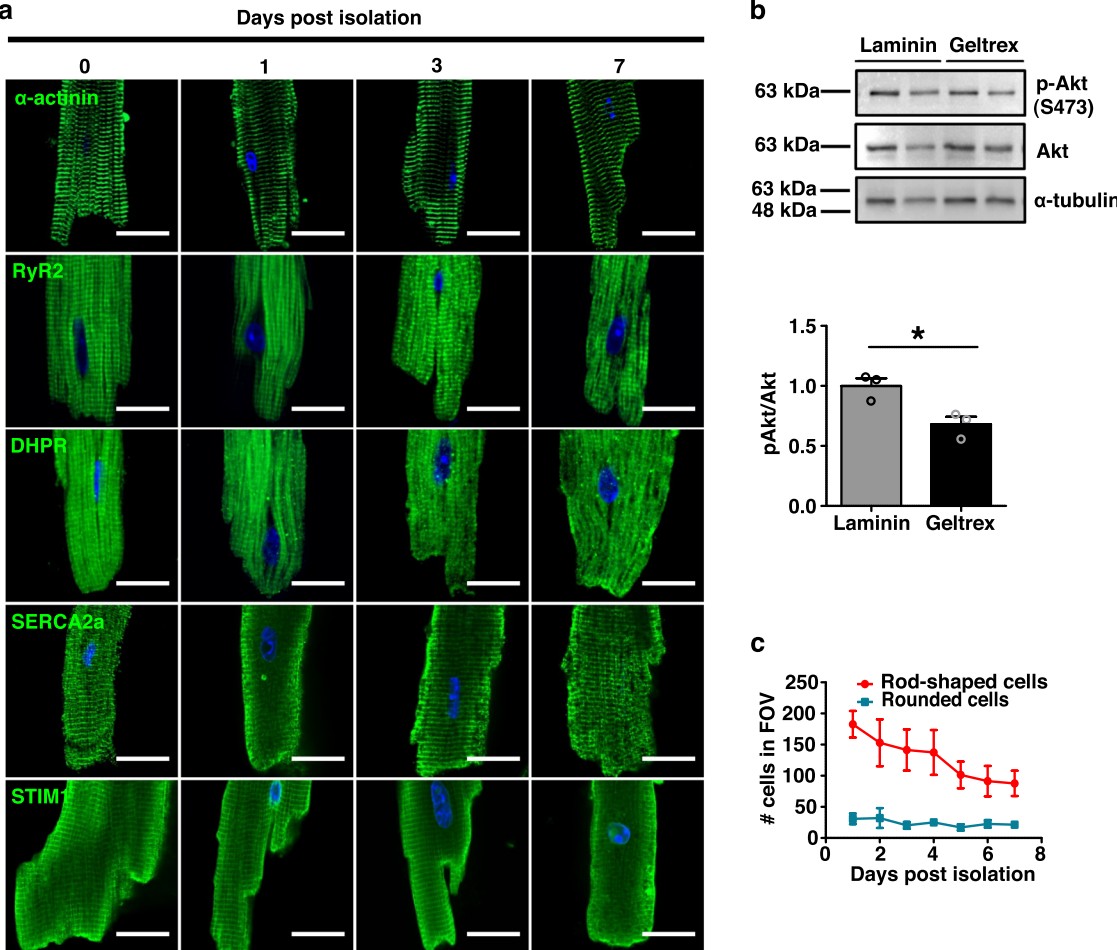

**Fig. 1 Isolated adult murine cardiomyocytes (aCMs) cultured on Geltrex and treated with blebbistatin show sustained viability and retention of functional protein expression patterns up to 7 days post-isolation. a** Confocal analysis of aCMs of α-actinin and RyR2, DHPRα, SERCA2a, and STIM1 (green) up to one week in culture. Nuclei were visualized with Hoechst 33342 (blue). Scale bars equal 20 μm. **b** Immunoblots for Akt, pAkt, and α-tubulin of aCM lysates after 24 h culture on Geltrex-coated surfaces ($N = 3$, $p = 0.023$). **c** Cell counting revealed the survival of viable rod-shaped (red) relative to rounded aCMs (blue) ($N = 3$). All data expressed as mean ± SEM; $N$ denotes biological replicates; significance indicated by *($p < 0.05$).

becoming significantly reduced by d3, followed by a recovery to levels approaching the initial isolation time values by d7. The decrease in $Ca^{2+}$ transient amplitude, with the amplitude most reduced at d3, coincided with the most prolonged transient rise and decay times. The apparent stabilization and return toward initial d0 levels by d7 would suggest acclimatization to culture. The decreased regularity in DHPR expression throughout the cell suggested that t-tubule organization was somewhat disrupted after extended culture (Fig. 1a). Since the L-type $Ca^{2+}$ current is important to induce SR $Ca^{2+}$ release in aCMs, aberrant DHPR expression corresponding to disrupted t-tubules may contribute to the altered $Ca^{2+}$ influx; disorganization of t-tubules may also impair the bulk of $Ca^{2+}$ efflux mediated by the $Na^+/Ca^{2+}$ exchanger although further investigation would be needed to characterize fully the changing electrophysiology of the aCM in culture. Similar changes to STIM1 expression patterns over time suggest that membrane and t-tubule physiology continues to adapt in vitro.

The rate of CellROX-active ROS production through live cell imaging increased steadily in culture to $149 \pm 10\%$ of baseline at d7 ($p < 0.0001$; Fig. 3c). This increase coincided with a significant decrease in FCCP-uncoupled maximal oxygen consumption rate ($117 \pm 12$ pmol $min^{-1}$ mg $protein^{-1}$ at d0 vs. $68 \pm 11$ pmol $min^{-1}$ mg $protein^{-1}$ at d7 $p < 0.0001$, Fig. 3d), suggesting growing metabolic inefficiency over increased culture time. However, increased maximal to basal OCR ratio (4.3 vs. 3.0) was noted at d1 compared to standard Langendorff isolation and culture methods[16], suggesting increased spare respirometric capacity early in culture with this protocol.

**Recapitulation of myocardial mechanics**. Traction force microscopy (TFM) has been carried out in many cell types, including neonatal and iPSC-derived cardiomyocytes[17]. However, its use in primary isolated aCMs has been hampered by poor cell attachment and functionality post-isolation. With the improvement of cardiomyocyte cell isolation with our protocol, we were able to apply widefield TFM techniques to aCMs to measure auxotonic contractions in a mechanically relevant environment (Supplementary Fig. 2). Spontaneous cell-associated total stresses were used to calculate single-cell contractile forces of 0.6–3.3 μN $cell^{-1}$. Cells plated on 2 kPa gels produced significantly lower peak forces than cells on gels of 11 kPa (Fig. 3e); this has previously been observed in neonatal CMs over extended culture periods[18,19]. Furthermore, given the speed of various mechanotransduction pathways in CMs[6,20,21], it is not surprising that these effects can be observed in an acute setting. When electrically paced, cells showed a positive force-frequency response, peaking

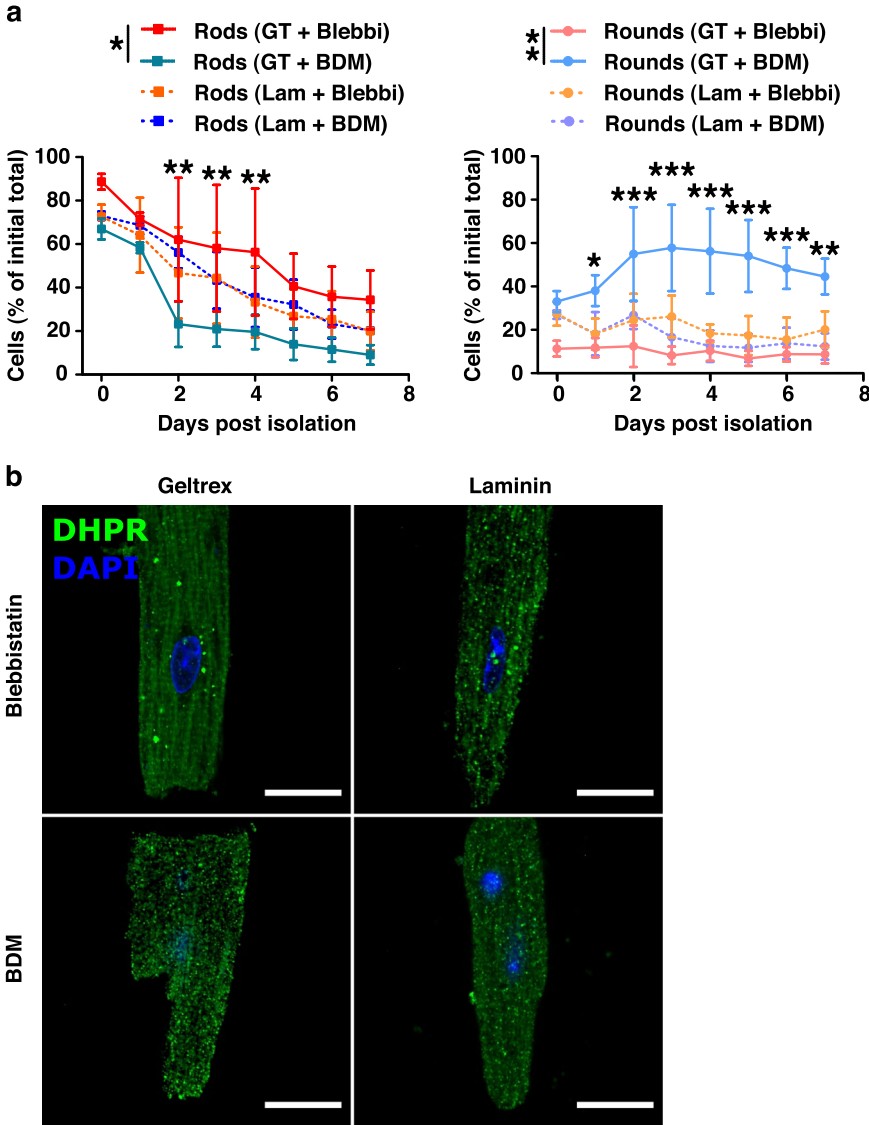

**Fig. 2 Adult cardiomyocyte cultures using a combination of Geltrex or laminin and the contractile inhibitor blebbistatin or butanedione monoxime over 1 week-post isolation. a** The combination of Geltrex (GT; solid lines) and blebbistatin (Blebbi; hot colors) produced a high ratio of rod-shaped myocytes (squares) to rounded (circles) cells compared to other component combinations using more conventional culture conditions including butanedione monoxime (BDM; cool colors) or laminin-521 (Lam; dotted lines). **b** Confocal imaging of DHPRα expression after 7 days in isolation. Striations and fewer puncta were seen in cultures with GT and blebbistatin than other combinations; scale bars represent 20 μm. All data expressed as mean ± SEM; $N = 3$ biological replicates except for the GT + BDM group ($N = 4$) and significant differences from the GT + blebbistatin culture treatment are indicated by *($p < 0.05$), **($p < 0.01$) and ***($p < 0.001$) for both main effects and individual timepoints.

between 2.5–4 Hz and 180% of baseline 1 Hz force (Fig. 3f) and approximating a quadratic curve fitting. The resulting Bowditch curve was significantly steeper and left-shifted compared to in vivo measurements of mice[22], ostensibly due to the effects of isolation and culture. Similarly, cells failed to reliably pace at field stimulation rates above 7 Hz (results not shown) while murine myocardium can pace in vivo to ~14 Hz. To assess the retention of functional aspects of the cells, we challenged them with epinephrine, a known beta-adrenergic agonist that increases contractility force. We determined that these cells clearly responded to epinephrine, and showed an increased peak contractile force of 68–96% increased force (Fig. 3g). Transmission electron microscopy of cells at d1 (Fig. 3h) revealed preserved sarcomeric structures and mitochondrial networks consistent with a healthy adult myocyte.

**Functional manipulations of cells in vitro.** Finally, we demonstrated the ability to modulate aCM function entirely in vitro (Fig. 4). Using myocytes isolated from phospholamban (PLN)-knockout CD1 mice, we used lentiviral transfection to transduce flag-tagged WT-PLN and a human pathogenic variant, R9C-PLN. Immunoblots (Fig. 4a) and confocal imaging (Fig. 4b) at d2 showed clear FLAG-PLN expression for both constructs. Furthermore, TFM analysis revealed anticipated differences between peak contractile forces ($2.45 \pm 0.22$, $1.44 \pm 0.11$ and $0.72 \pm 0.12$ μN cell$^{-1}$ for PLN KO, WT-PLN, and R9C-PLN, respectively; $p < 0.001$), likely as a direct consequence of the known effects of the PLN constructs through SERCA2a regulation (Fig. 4c). These experiments clearly highlight the feasibility of utilizing these aCMs for biochemical and functional assays in genetic studies. In parallel experiments, we also demonstrated

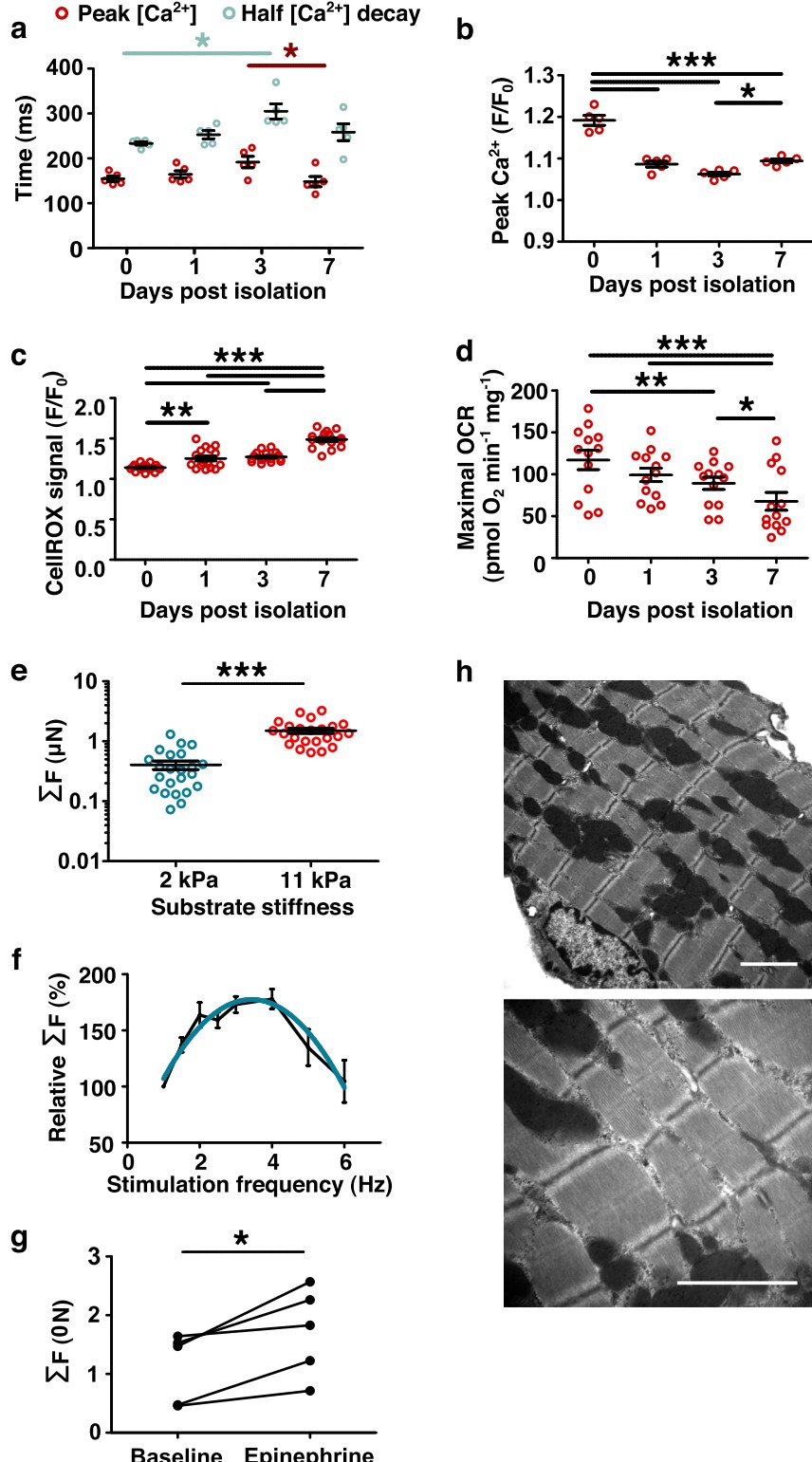

**Fig. 3 Critical hallmarks of aCM function are retained up to 7 days in culture. a** $Ca^{2+}$ transients of isolated myocytes were recorded to measure time to peak amplitude and time to decay for up 7 days ($N = 5$). **b** Peak $Ca^{2+}$ amplitudes were determined up to 7 days in culture ($N = 5$). **c** Normalized intracellular reactive oxygen species stain (CellROX) signal was measured over time in culture ($N = 3$). **d** Maximal FCCP-uncoupled oxygen consumption rate (OCR) over culture period ($N = 3$). **e** Peak contractile forces by spontaneously contracting aCMs were measured on TFM substrates of 2 kPa and 11 kPa stiffness ($N = 20$ for each treatment). **f** Electrically stimulated aCMs show a physiological (Bowditch)-resembling force-frequency relationship that scales relative to the absolute force of the cell at $f = 1$ Hz ($N = 5$), quadratic fit $R^2 = 0.61$. **g** Peak contractile force measurements measured by TFM in the absence and presence of epinephrine (5 nmol $L^{-1}$) ($N = 5$ paired measurements). **h** Transmission electron microscopy (TEM) at two levels of magnification; scale bars 2 μm. All data expressed as mean ± SEM; $N$ denotes biological replicates; significance indicated by *($p < 0.05$), **($p < 0.01$), and ***($p < 0.001$).

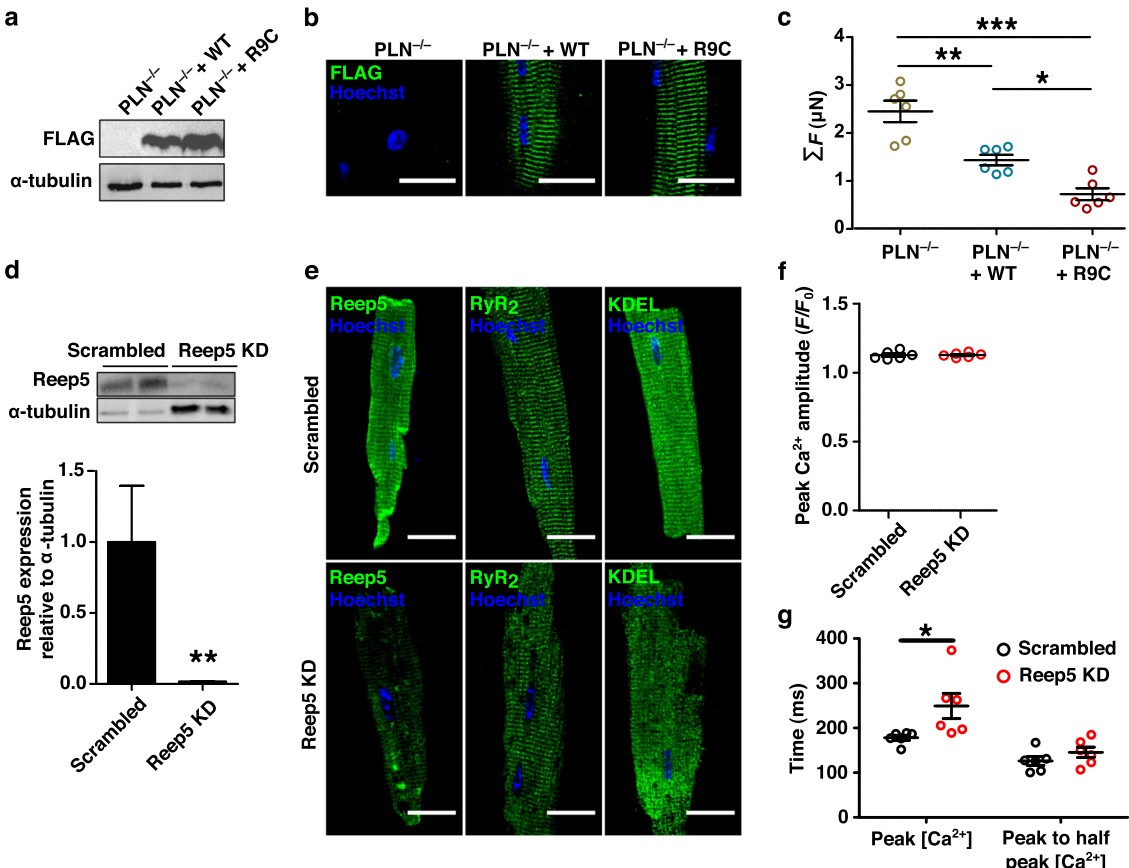

**Fig. 4 Isolated murine aCMs can be genetically manipulated in vitro to functional effect. a** Myocytes isolated from a PLN-null mouse were transfected using lentiviral vectors expressing wildtype (WT) and human pathogenic R9C phospholamban (PLN) flag tagged cDNAs. Immunoblots to anti-flag and α-tubulin. **b** Confocal images of aCMs stained for FLAG expression post lentiviral transfection for PLN. Scale bars represent 20 μm. **c** Traction force microscopy of aCMs post transfection. **d** Myocytes isolated from a WT mouse were transfected using AAV9 vectors encoding Reep5 shRNA sequence and assessed by immunoblots against REEP5 and tubulin. Reep5 KD treatment were loaded at higher volume to produce quantifiable bands. **e** Confocal images of aCMs confirm shRNA knockdown of Reep5 and resulting disorganization of SR (RyR2) and ER (KDEL motif) compared to scrambled shRNA control; scale bars represent 20 μm. **f** Peak $Ca^{2+}$ transient amplitude is not significantly affected by Reep5 KD ($N = 6$). **g** Time to peak $Ca^{2+}$ transient amplitude is significantly higher after Reep5 KD, but transient decay time to 50% is not affected. All data expressed as mean ± SEM; $N$ denotes biological replicates; significance indicated by *($p < 0.05$), **($p < 0.01$), and ***($p < 0.001$).

knockdown of gene expression in these cells, further emphasising the utility of the cellular isolation methodology. Specifically, as the SR adaptor protein Reep5 is known to be essential in maintaining cardiac function[23], AAV9-mediated Reep5 shRNA knockdown (Fig. 4d) was performed and cells were analyzed at d3 or d4. Transduced cardiomyocytes showed disorganized SR/ER morphology (Fig. 4e) and increased spontaneous $Ca^{2+}$ efflux time relative to treatment with scrambled control shRNA (Fig. 4f). Similarly, ER stress induced by 24 h tunicamycin treatment (5 μmol L$^{-1}$) resulted in extensive t-tubular disorganization and vacuolization as assessed by dSTORM super-resolution imaging (Fig. 5). These sample workflows can be adapted for in vitro characterization of pathways and mechanisms of interest in early disease progression, similar to those already performed in primary cultures of other tissues.

## Conclusions

We have shown an important advantage of utilizing Geltrex and blebbistatin to culture and analyze adult cardiomyocytes enabling an unprecedented level of experimentation in aCMs. However, there are some limitations to the widespread utility of this protocol. The use of blebbistatin as an inhibitor at the point of contraction likely prevents dynamic mechanical transduction in

the cell without the off-target effects induced by BDM. Furthermore, our efforts to culture cells in confluent monolayers or 3D tissues were impaired by lack of alignment and less than ideal adhesion to the surface. Without intercellular contact, lack of regular impulse propagation and syncytial passage of intracellular molecules also represents a point of divergence from in vivo homeostasis. Finally, restriction of dissolved oxygen levels to those approximating arterial partial pressures would likely improve myocyte metabolism and/or limit ROS exposure and further improve cellular culture conditions[24]. These important distinctions may represent potential paths for future improvement in the culture system to enable longer-term culture for advanced experiments and pharmaceutical testing.

Altogether, this study demonstrates an innovation to primary murine aCM cultures with increased cell longevity and durable physiological function in vitro, notably allowing non-acute experimental pharmacological and genetic manipulations in aCM culture for the first time to our knowledge. By combining the physiological benefits of blebbistatin and Geltrex, the loss of aCM phenotypes and cell death is reduced, with aCM hallmarks retained over at least 7 days of culture. Importantly, we have demonstrated the ability to model pathology through both genetic modulation and small-molecule administration due to the enhanced timescales and experimental flexibility afforded by

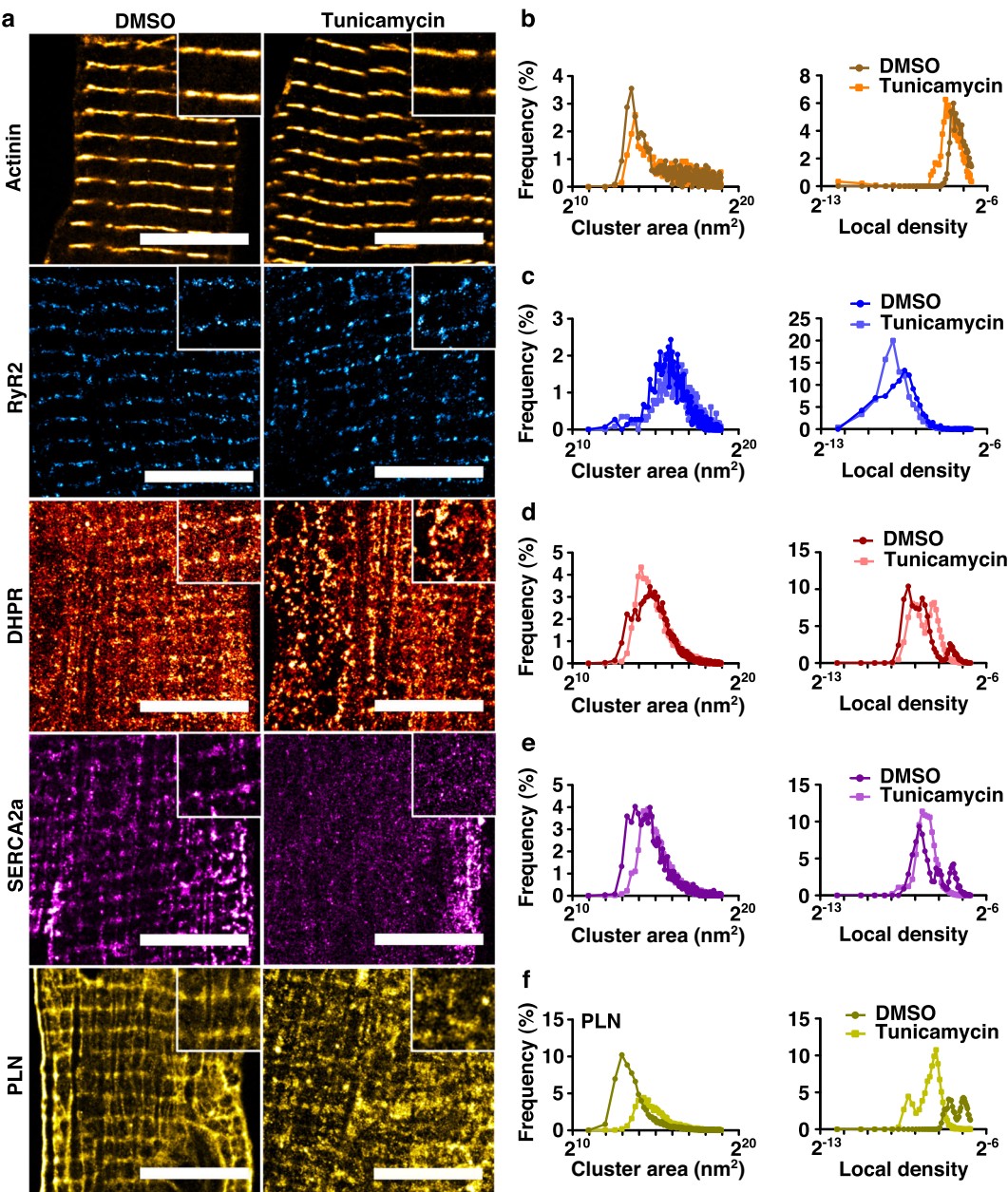

**Fig. 5 Super-resolution (dSTORM) microscopy reveals that aCMs are sensitive to chemical perturbations in vitro. a** dSTORM imaging of aCM to visualize α-actinin, RyR2, DHPR, SERCA2a, and PLN in the absence or presence of tunicamycin-induced protein folding stress (5 μmol L$^{-1}$ for 24 h vs. a DMSO sham). Scale bars represent 10 μm, magnified inset images are 4 × 4 μm. **b** Voronoi tessellation cluster area and density analysis of dSTORM-imaged aCMs ($N = 3$) for α-actinin (**b**), RyR2 (**c**), DHPR (**d**), SERCA2A (**e**), and PLN (**f**).

these methods. Finally, to our knowledge, we also describe the first use of TFM for physiological, treatment-sensitive force measurement of primary aCM auxotonic contraction. Together, these tools will allow for the application of experimental treatments and measurements previously restricted to in vivo settings to an in vitro model, with the mechanistic control afforded by the single-cell scale.

## Methods

**Ethical statement.** All studies were approved by the University of Toronto Animal Care Committee and conducted under Institutional Animal Care Guidelines.

**Reagents, media, and consumables.** Perfusion and EDTA buffers were prepared as previously described[2], except that they contained 15 μmol L$^{-1}$ (S)-(−)-blebbistatin (Toronto Research Chemicals, Toronto; ON). EDTA, taurine, and HEPES were supplied by BioShop Inc. (Burlington, ON). M199 (Wisent Inc., Saint-Jean-Baptiste,

QC) pH 7.6 was supplemented with added 100X CD lipid, 100X insulin-transferrin-selenium supplement (Life Technologies), and 100X penicillin/streptomycin to 1X each. All other reagents unless mentioned were supplied by Sigma-Aldrich (St. Louis, MI).

**Isolation of primary murine adult cardiomyocytes.** The surgical and perfusion methods closely followed those of a recent optimization of a Langendorff-free preparation[2]; we encourage readers to refer to Ackers-Johnson et al. for detailed considerations in surgical methods and culture optimization. Complete media recipes are provided in Supplementary Table 1. Briefly, male CD1 mice of 8 weeks or older were euthanized by open drop exposure to isoflurane followed by cervical transection. The chest cavity was opened, the descending aorta severed, and 7 mL of EDTA buffer containing 15 μmol L$^{-1}$ blebbistatin (Toronto Research Chemicals, Toronto, ON) was injected into the right ventricle. The heart was hemostatically clamped at the ascending aorta, excised from the chest cavity, and placed into a dish of fresh EDTA buffer containing 15 μmol L$^{-1}$ blebbistatin while 9 mL of the same buffer was injected slowly into the apex of the left ventricle. After the heart was cleared of blood, it was moved to a dish of perfusion buffer with 15 μmol L$^{-1}$

blebbistatin, and injected with 3 mL of fresh 15 μmol L$^{-1}$ blebbistatin through the same hole previously used in the LV. Finally, the heart was moved to a dish containing 475 U mL$^{-1}$ collagenase type II (Worthington Biochemical Corporation, Lakewood NJ) in perfusion buffer with 15 μmol L$^{-1}$ blebbistatin, of which 20 mL more was injected through the existing LV opening. We found that the use of 27 G, ½ length needles minimized mechanical damage to the heart, allowing for the maintenance of pressure during perfusion and optimal coronary circulation of collagenase and, thus, digestion of the myocardium. Additionally, we observed batches of higher specific activity (IU/mg) collagenase type II to be more amenable to cell survival, even at the same final activity concentration.

At this point, tissue was minced in 3 mL of fresh collagenase buffer with forceps, and gently triturated with a wide-bore 1 mL pipette. Collagenase activity was inhibited with the addition of 3 mL of perfusion buffer with blebbistatin and 10% FBS (Wisent)). The isolate was then passed through a 70 μm strainer and rinsed with 3 mL additional stop buffer. The filtrate was divided between two 15 mL Falcon tubes which were left standing upright for 15 min. The rod-shaped viable cardiomyocytes gravity-settled to form a deep red pellet, while rounded, nonviable aCMs and other cell types remained in suspension. The use of 2 tubes prevented oxygen or nutrient gradients forming in the cell pellets, while the use of steep-walled 15 mL Falcon tubes allowed for the best recovery of the pellet over successive washes. The supernatant was removed carefully, and the cells resuspended in a mixture of 75% perfusion buffer and 25% culture media, containing 15 μmol L$^{-1}$ blebbistatin. Cells were allowed to settle 15 min, and the process repeated two more times with mixtures of 50%:50% and 25%:75% perfusion buffer:culture media, respectively, all containing 15 μmol L$^{-1}$ blebbistatin. The final cell pellet was resuspended in culture medium containing 5% FBS and 15 μmol L$^{-1}$ blebbistatin, which was then plated for culture in the required format. After plating for 3 h, dishes were gently washed and culture media was replaced containing 15 μmol L$^{-1}$ blebbistatin (no FBS) to avoid serum toxicity. Cells were then cultured typically up to 7 days post-isolation for functional analyses, although we noted substantial CM survival 3 weeks post-isolation without full phenotypic characterization.

**Cell culture and treatments.** Cells were cultured on 14 mm glass coverslips coated with 250 μL Geltrex diluted 1:100 in M199 at c.a. 100,000 cells/coverslip in 35 mm dishes (P35GCOL-1.5-14-C, MatTek, Ashland, MA) for immunofluorescence, ROS, and Ca$^{2+}$ imaging. For immunoblotting, cells were plated in 6-well tissue culture polystyrene (TCPS) plates at ~1 × 10$^6$ cells well$^{-1}$ with plates coated with 1.5 mL Geltrex diluted 1:100 in M199 or laminin-111 (Trevigen Inc., Gaithersburg, MD, USA) at 5 μg mL$^{-1}$ for direct comparisons of Geltrex vs. laminin. For Agilent Seahorse respirometric characterization, cells were plated at ~ 250,000 cells well$^{-1}$ in 24-well Seahorse plates coated with 50 μL Geltrex diluted 1:100 in M199.

For viral transfection, aCMs 4 h post-plating, pretreated with 10 μg mL$^{-1}$ polybrene (TR-1003, Millipore-Sigma), were treated for a further 21 h with a lentiviral vector containing PLN-WT or PLN-R9C, or an AAV9 vector containing Reep5 or scrambled shRNA, prepared as described previously[25]. For the PLN experiment, cells were fixed or subjected to TFM analysis 24 h after viral introduction. For the Reep5 experiment, cells were incubated a further 24 h after the vector was removed before live Ca$^{2+}$ imaging or fixation. For tunicamycin treatments, cells 4 h post-plating were incubated 24 h in 5 μmol L$^{-1}$ tunicamycin from *Streptomyces* sp. (T7765, Millipore-Sigma) from a 1 mg mL$^{-1}$ DMSO stock and compared to a vehicle sham.

**Antibodies.** Rabbit polyclonal anti-Akt antibody (1:1000 for IB, 9272; Cell Signaling Technology), rabbit polyclonal anti-pAkt-Ser473 antibody (1:1000 for IB, 9271; Cell Signaling Technology), rabbit polyclonal anti-α-tubulin antibody (1:1000 for IB, 2144, Cell Signaling Technology), mouse monoclonal anti-α-actinin antibody (1:400 for IF, A7811; Sigma-Aldrich), mouse monoclonal anti-ryanodine receptor antibody (1:100 for IF, ab2827; Abcam), mouse monoclonal anti-dihydropyridine receptor (DHPR) antibody (1:800 for IF, ab2864; Abcam), mouse monoclonal anti-sarco(endo)plasmic reticulum Ca$^{2+}$-ATPase (SERCA) 2a antibody (1:200 for IF, MA3-919; Thermo-Fisher), rabbit polyclonal anti-STIM1 antibody (1:200, PA1-46217; Thermo-Fisher), mouse monoclonal PLN [2D12] antibody (1:500 for IF, 1:1000 for WB; ab2865, Abcam), mouse monoclonal FLAG antibody (1:500 for IF, 1:1000 for WB, F1804; Millipore-Sigma), mouse monoclonal anti-Reep5 antibody (1:500 for IF, 1:1000 for WB, 14643-1-AP; Proteintech), and rabbit monoclonal anti-KDEL antibody (1:250 for IF, ab176333; Abcam) were used in this study. Goat anti-rabbit Alexa Fluor 488 secondary antibodies (nos. A-11034 and A-11011; Molecular Probes) were used at 1:800 dilution.

**Immunoblotting.** For Akt signaling analysis and genetic experiments, protein lysates from aCMs at high density and plated in a single well of a 6-well TCPS plate (0.5–1 heart well$^{-1}$) were harvested in radioimmunoprecipitation assay buffer (RIPA, 50 mM Tris-HCl; pH 7.4, 1% NP-40, 0.5% sodium deoxycholate, 0.1% SDS, 150 mM NaCl, 2 mM EDTA, 1X cOmplete Mini protease inhibitor cocktail (4693159001, Roche). For PLN expression analysis, cells were lysed in lysis buffer (8 mol L$^{-1}$ urea, 10% (v/v) glycerol, 20% (w/v) SDS, 1 mol L$^{-1}$ dithiothreitol, 1.5 mol L$^{-1}$ Tris-HCl, pH 6.8, 1X cOmplete™ Mini protease inhibitor cocktail

(4693159001, Roche)) with an 18-gauge needle. Lysates were centrifuged at 15,000 × g for 15 min at 4 °C.

SDS-soluble supernatants were added to 2X loading buffer and subjected to SDS-PAGE in a 12% polyacrylamide gel with 6% stacking gel at 100 V for 20 min, then 120 V for 1 h. Semi-dry transfer to a PVDF membrane occurred at 70 V for 1 h. Membranes were blocked in 5% BSA in TBS + 0.05% Tween-20 for 1 h at room temperature, then incubated overnight at 4 °C in primary anti-Akt, anti-pAkt (Ser473), anti-Reep5, anti-FLAG, or anti-α-tubulin antibodies (described above), followed by secondary antibodies (1:2500 dilution) for 1 h at room temperature. ECL detection was performed with a ChemiDoc™ Touch (Bio-Rad Laboratories, Hercules, CA).

**Confocal microscopy.** Cultured cells were fixed with 4% paraformaldehyde for 10 min on ice, followed by 90% ice-cold methanol for 10 min. Next, cells were incubated with permeabilization buffer (0.5% Triton X-100, 0.2% Tween-20 in PBS) for 30 min at 4 °C. Blocking buffer (5% FBS in 0.1% Triton-X-100 in PBS) was then added and incubated for 30 min at room temperature. Cells were incubated with primary antibodies (SERCA2a—1:500, PLN—1:1000, RyR2—1:1000, DHPRα —1:700) in blocking buffer overnight at 4 °C, and fluorophore-conjugated secondary antibody staining (Alexa 488; Molecular Probes) was performed at room temperature for 1 h in the dark. Nuclear counterstaining was performed using 1 μg ml$^{-1}$ Hoechst 33342 (no. 4082; Cell Signaling) at room temperature for 15 min in the dark. Cells were imaged using a Zeiss spinning-disk confocal microscope and processed using Zen Pro software (Zeiss).

**Traction force microscopy.** TFM analysis in many cell types is often conducted by confocal microscopy, where a detergent is used to solubilize a cell to relieve its traction stress on gel. There, confocal microscopy allows for the imaging of only a single layer of gel. However, to characterize physiological CM contractions, widefield fluorescent microscopy is needed for temporal resolution. Therefore, TFM beads must be limited only to the surface of the gel to prevent under-estimation of cell tractions by capturing beads in lower planes with smaller displacements. To this end, 18 mm circular coverslips were coated in a suspension of 500 nm red carboxylated FluoSpheres (580 nm excitation and 605 nm emission maxima; F8812, Thermo-Fisher) diluted 1:300 (v:v) in 100% ethanol, and slowly dried in a closed 12-well polystyrene culture plate to prevent heterogenous deposition of fluospheres. Matching coverslips were washed in 1 mol L$^{-1}$ NaOH, washed with deionized water, dried, coated with (3-aminopropyl) triethoxysilane (APTES) for 10 min, then rinsed in deionized water again. An 11 kPa poly-acrylamide (PA) solution (715 μL 50 mM HEPES pH 7.4, 150 μL 2% bis-acrylamide (Bio-Rad), 125 μL 40% acrylamide, 5 μL 10% ammonium persulfate, and 1 μL TEMED) was prepared, and 80 μL immediately spread on the silanized coverslip. 2 kPa PA gels were prepared similarly, except the volumes of HEPES, 2% bis-acry-lamide, and acrylamide were 815, 40, and 137.5 μL, respectively. The coverslip prepared with FluoSpheres was then gently floated on the solution, and the solution left to polymerize 30 min. The top coverslip was gently removed, leaving behind its FluoSphere coating at the surface of the gel. The gel remained conjugated to the APTES-functionalized bottom coverslip, which was placed in one well of a 12-well plate. Gels were washed 3× with PBS. Protein conjugation to the surface of the gel was accomplished as previously described[26]. Briefly, N-sulfosuccinimidyl-6-(4'-azido-2'-nitrophenylamino) hexanoate (sulfo-SANPAH, CovaChem, Loves Park, IL) was solubilized in DMSO (0.25% final concentration) before diluting to 500 mmol L$^{-1}$ in 50 mmol L$^{-1}$ HEPES pH 7.4. The solution (2 mL) was added immediately to the well containing the gel, and exposed to 365 nm UV light for 10 min. This process was repeated with a fresh aliquot of sulfo-SANPAH solution. Gels were rinsed three times with 50 mM HEPES. Gels were then incubated overnight at 4 °C in 1 mL 1:50 Geltrex in PBS for covalent attachment. Gels were rinsed 3x with PBS before plated with cells.

aCMs were plated on the gels as described above. For contractile analysis, wells were rinsed and then incubated with blebbistatin-free culture media for 5 min at 37 °C. Spontaneous contractions as visualized by displacement of the FluoSpheres were then recorded on an IX71 inverted widefield fluorescent microscope (Olympus Corporation, Tokyo, Japan) with a Texas Red filter cube at timelapse series with exposures of 55 ms. Brightfield images of the contracted cell were also taken for integration of the strain vectors (described below). For force-frequency curves, cells on 11 kPa gels were field-stimulated using two carbon electrodes of c.a. 1.5 cm length and 1 cm distance, soldered to copper leads that were then insulated with silicone rubber. For force-frequency curves, an S48 physiological monophasic square wave stimulator (Grass Technologies, Warwick, RI) was used to pace contraction from 1 to 6 Hz at 50 V, 5 ms duration.

Frames of peak contraction and relaxation were analyzed using a particle image velocimetry (PIV) plugin for ImageJ (v1.51j8)[27]. Interrogation windows of 64 × 64 pixels in 128 × 128 pixel search windows with a 0.60 correlation threshold were used to generate a displacement field, which was then used as the basis for Fourier transform traction cytometry (FTTC) as calculated by a separate ImageJ plugin[27]. A Poisson's ratio of 0.48, Young's modulus of 11.0 kPa or 2.0 kPa, and a unitless regularization parameter (λ) of $4.7 \times 10^{-10}$ were used. The resulting stress matrix was integrated within the borders of the cell as captured by brightfield microscopy using a custom Matlab 2018a script. This sum was multiplied by the area of the cell to produce a total force scalar, and divided by 2 assuming both uniaxial force

production by the cell, and null net force production given a complete reversion to the pre-contraction state. Final force values were expressed in terms of whole-cell peak force. Representative cells across $N = 5$ animals were measured for the 2 kPa vs. 11 kPa comparison and Bowditch curve. Duplicate separately-treated wells (one cell per well) for $N = 3$ animals were assessed for each treatment of the PLN-knockout experiment. Measurements were taken at d0 for stiffness sensitivity, force-frequency, and epinephrine response measurements, and at d2 for the PLN-knockout experiment for consistency with imaging and to ensure adequate response time.

**Reactive oxygen species (ROS) staining.** aCMs were incubated 30 min in culture medium with blebbistatin, containing 5 μmol $L^{-1}$ CellROX Green (Thermo-Fisher) and counterstained 5 min with 1:1000 Hoechst 33342, both according to the manufacturer directions. Cells were washed in fresh media and immediately imaged with a ×40 objective on an inverted IX71 widefield microscope and using MicroManager acquisition software with exposures of 200 ms with a FITC filter and 50 ms with a DAPI filter. Fluorescent intensity per unit area was normalized to an equivalent area of adjacent background using ImageJ.

**Respirometry.** Freshly isolated aCMs were plated in 24-well Seahorse XFe™ (Agilent Technologies Inc, Santa Clara, CA, USA) plates and cultured as described above for timepoint analysis. Mitochondrial respiration was assessed 0, 1, 3, and 7 days post-isolation in a Seahorse XFe24 bioanalyzer. Cells were incubated in DMEM XF assay media (#102353-100, Agilent) supplemented with 5 mmol $L^{-1}$ glucose, 1 mmol $L^{-1}$ pyruvate, and 2 mmol $L^{-1}$ glutamine at 37 °C in a $CO_2$-free incubator for 1 h prior to assay. Injector ports were loaded to provide final concentrations of 1 μmol $L^{-1}$ oligomycin, 0.5 μmol $L^{-1}$ FCCP, and 1 μmol $L^{-1}$ rotenone and 2 μmol $L^{-1}$ antimycin-A together, respectively according to the mitochondrial stress test protocol provided by Agilent. Maximal respiration was calculated and normalized to total protein concentration as measured by a Bradford assay. Optimal oligomycin and FCCP concentrations were determined by titration.

**$Ca^{2+}$ imaging.** Fluo-4 AM (Thermo-Fisher Scientific), reconstituted in DMSO and frozen at −20 °C, was incubated in the dark with cells at 37 °C for 30 min at a final concentration of 5 μmol $L^{-1}$. Spontaneous $Ca^{2+}$ waves were recorded using an IX71 widefield microscope and MicroManager acquisition software at 55 ms exposures with a FITC filter and expressed normalized to baseline fluorescence ($F/F_0$) using ImageJ processing.

**Transmission electron microscopy.** Isolated aCMs were fixed in 2.5% glutaraldehyde in 0.1 mol $L^{-1}$ phosphate buffer at 4 °C overnight. Samples were postfixed in 1% osmium tetroxide buffer and processed through a series of ethanol dehydrations, and embedded in Quetol-Spurr resin. Sections of 90–100 nm were cut and stained with 1% uranyl acetate and 3% lead citrate and imaged at ×20,000 magnification using a Hitachi TE microscope at the Department of Pathology, St. Michael's Hospital (Toronto, Canada).

**dSTORM super-resolution imaging.** Direct stochastic optical reconstruction microscopy was carried out as described previously[28]. Briefly, stochastic photo-switching of immunostained samples was initiated with a buffer containing 50 mmol $L^{-1}$ 2-mercaptoethylamine (M9768, Sigma-Aldrich), 40 μg m$L^{-1}$ catalase (C3155, Sigma-Aldrich), 500 μg m$L^{-1}$ glucose oxidase (G7141, Sigma-Aldrich), 50% (w/v) D-glucose (Sigma-Aldrich), in PBS pH 7.4. A 643 nm laser set at 20 mW was used to inactivate the AlexaFluor 647 fluorophore into an off-state prior to stochastic reactivation over the acquisition period. A super-resolved composite of 10,000 images acquired over a period of 300 s at 30 ms exposures was then reconstructed using the ThunderSTORM v1.3 ImageJ plugin, using a linear least square localization method. Coordinates of single emitters were filtered based on localization precision and photon count to discard electronic noise (0 nm < localization precision < 7 nm) and sample noise (localization precision >60 nm).

**Statistics and reproducibility.** All experiments were performed at least 3 times. Statistical analysis was conducted with Prism 5 (GraphPad Software Inc.), except for respirometric analyses which were analyzed by JMP 11 (SAS Institute, Caly, NC, USA). All treatments were tested using the D'Agostino and Pearson omnibus normality test before comparison with an unpaired t-test, except for the epinephrine experiment which was analyzed by paired t-test. The comparison between treatment combinations was analyzed by 2-way repeated measures ANOVA for all groups, followed by post-hoc 2-way repeated measures ANOVA by pairs against Geltrex/blebbistatin with Bonferroni correction; individual treatments were compared between significantly different treatments using the Bonferroni post-test. Respirometric timepoint data was assessed by 1-way split-plot ANOVA, followed by Tukey–Kramer HSD. Differences between timepoints and PLN cDNA knock-ins were assessed by 1-way ANOVA followed by Tukey's LSD, except for Seahorse experiments which were assessed by 1-way repeated measures ANOVA followed by Tukey's LSD. The contractile Bowditch curve was fitted with a quadratic regression

($a = -11.91$, $b = 81.80$, $c = 37.08$, $R^2 = 0.605$). Differences were considered significant at $p < 0.05$.

**Reporting summary.** Further information on research design is available in the Nature Research Reporting Summary linked to this article.

## Data availability

The datasets generated during this study are available from the corresponding author (Anthony Gramolini). Source data for all graphs shown along with uncropped immunoblots are included in the Supplementary materials for this study.

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

## Acknowledgements

The authors thank Dr. Eric Strohm and Dorrin Zarrin-Khat for expert experimental assistance. The custom Matlab script was written and provided by Richard Tam. PLN$^{-/-}$ mice were a kind gift from Dr. Evangelia Kranias (University of Cincinnati). This study was funded by Canadian Institutes of Health Research (CIHR) operating grants (MOP-123320 and GPG-102166), CIHR/Natural Science and Engineering Research Council of Canada (NSERC) Collaborative Health Research Project grants (CHRPJ 478473-15/CPG-140194 and CHRPJ 508366-17/CPG-151946), Heart and Stroke Foundation of Ontario (T-6281); NSERC (RGPIN-2016-05618 and RGPIN-2015-043); and the Translational Biology and Engineering Program (TBEP). NIC was supported by an NSERC Vanier Canada Graduate Scholarship, an Ontario Graduate Scholarship, and a C. David Naylor Fellowship Endowed by a gift of the Arthur L. Irving Foundation. S.-H.L. was supported by an NSERC PGS-D, a Ted Rogers Centre for Heart Research Doctoral Fellowship, Peterborough Hunter Fellowship, and a Joe Connolly award. S.H.-L. and X.A. L. were supported by Ontario Graduate Scholarships. M.A.S. was supported by a Ted Rogers Centre for Heart Research Postdoctoral Fellowship.

## Author contributions

N.I.C., S.-H.L., S.H.-L., C.A.S., and A.O.G. conceived the study. N.I.C., S.-H.L., S.H.-L., X.A.L., M.A.S., and A.D. conducted experiments and analyzed data. N.I.C. wrote the paper. N.I.C., S.-H.L., S.H.-L., X.A.L., M.A.S., A.D., C.M.Y., M.H., C.A.S., and A.O.G. interpreted findings and edited the paper.

## Competing interests

The authors declare no competing interests.
