## [Peer Review File · Communications Biology]

Reviewers' comments:

Reviewer #1 (Remarks to the Author):

1. What are the major claims of the paper?

The authors present a combinatory approach to culturing harvested adult mouse cardiomyocytes (aCMs). Under traditional culture protocols, aCMs rapidly lose their phenotype (e.g., their rod-shaped structure) and undergo cell death. The authors show that with their protocol, aCMs remain viable and retain their phenotype for 7 days - enough time to do genetic manipulation and small molecule screening.

2. Are they novel and will they be of interest to others in the community and the wider field? If the conclusions are not original, it would be helpful if you could provide relevant references.

Most studies with aCMs are indeed acute studies because of aCM de-differentiate and die in the culture so rapidly. I believe this protocol could be very effective in changing the field.

3. Is the work convincing, and if not, what further evidence would be required to strengthen the conclusions?

The work is generally convincing but the results could be presented in a more direct way. For example,

3.1 the authors never show the % of viable rod-shaped (aka adult-like) cells under traditional culture conditions. This could be shown in Figure 1 a and c since Fig 1b already show a comparison of Akt activity in aCMs seeded on Laminin or Geltrex

3.2 Similarly, the new protocol features a combination of Geltrex and Blebbistatin but it is never shown what are the effect of cell death and de-differentiation of each component.

3.3 At various points, the authors reference past published reports on the use of Geltrex and Blebbistatin (or BDM) but never clarify the specific way their approach is different than those previous reports. What is the secret here? The combination? The specific concentrations?

3.4 Another point of confusion is at pag 3, line 53 - where the authors state that previous protocols could not preserve aCMs contractile structure past day 8. Unfortunately, the authors' data show aCMs contractile structure up until day 7, so it's unclear whether they can do better than earlier attempts in this respect. Can this be re-stated? Moreover, this would be much better illustrated with data showing aCMs de-differentiation in laminin-only (traditional) cultures.

3.5 The paragraph continues with a convincing demonstration that the new protocol is better at preserving ultrastructural elements, such as t-tubules, needed for calcium handling. This argument is further discussed convincingly in the following paragraph. Yet, the statements in pag 4, line 79-83 need to be revised. CICR can be triggered by T-Type calcium currents, not just L-type ones (change the word "necessary" with "important"). Moreover, a reference is needed to support the notion that Fluo-4 can be used to track the "delay" calcium fluxes associated with a switch from L- to T-type currents and the loss of t-tubules. In general, this section seems to invoke in an imprecise manner a lot more physiology that's needed to support the simple claim that the new protocol preserves calcium cycling in aCMs better than the old one.

3.6 The TFM data is very important, as it is possibly the first time this technique could be used on adult cells. At the same time, One technical question I could not answer in the Methods section is on which day the TFM experiments were performed, and how long the aCMs were kept on TFM substrates before experiments. TFM substrates do not feature GelTrex so it would be important to at least discuss the potential for de-differentiation if aCMs are kept on these substrates for too long.

3.7 Also on TFM, the authors compare contractile profiles at 2 and 11 kPa but the interesting comparison would have been between de-differentiated (fetal-like) and preserved aCMs. In fact, 2 kPa is the stiffness of fetal tissues and 11 kPa the stiffness of adult one. That is, do de-differentiated cells contract more strongly on 2 kPa tissues as shown for example by the Parker lab? Alternatively, and as before, I would limit the comparison to the 11 kPa substrates but focus on aCMs cultured with the new protocol and the ones cultured with the old protocol at 1 and 7 days after seeding.

3.8 Fig 3 and 4 showing how the new protocol enables important approaches on physiologically relevant adult CM is spot-on and made even more impressive by the use of novel techniques such as dSTORM.

4. On a more subjective note, do you feel that the paper will influence thinking in the field? Please feel free to raise any further questions and concerns about the paper.

I like the paper a lot and believe it could be very important for the field, as the types of important experiments described in Fig 3 and Fig 4 are currently possible only on cell lines, neonate primary CM, or stem cell-derived CM - all cell types that do not recapitulate a healthy, adult-like phenotype. At the same time, Fig 1 and 2 only show the good results obtained with the new protocol without providing a direct comparison with the bad outcomes of standard culture conditions or sub-optimal combinations of the two elements they propose. If a stronger basis is provided in Fig 1 and 2, Figures 3 and 4 will have an even bigger impact.

Reviewer: Francesco Pasqualini

Reviewer #2 (Remarks to the Author):

- While promising, maintaining aCM cultures for 7 days is a relatively short time span to claim "long-term" culture method. I would suggest that the authors extend the culture duration and characterization to at least 14 days, which will put them in line with drug screening assays and tissue function recapitulation experiments. This will also significantly enhance the impact of this work.

- The paper described a method to maintain a difficult cell culture in a more or less functional state, which will help the field where such culture times are necessary. The authors claim that this will help the field recapitulate tissue functions in vitro, but they offer very limited experiments to show this. I would suggest at least a one additional experiment where they use a drug with known effects to compare their culture method to current state of the art culture method.

- The statistical analysis, where needed is appropriate and convincing.

- The work would be possible to replicate given the level of detail provided in the paper and it would be of interest to others in the field.

I would recommend to accept with revisions

Reviewer #3 (Remarks to the Author):

These studies establish an improved method for culturing adult cardiomyocytes (aCM) from mice. The investigators introduce the use of Geltrex as an ECM substrate and the use of myosin ATPase inhibitor blebbistatin to reduce aCM contractility on the ECM. They demonstrate that the aCM maintain structure and function over a 7-day period and, during that window, the cells can be genetically and pharmacologically manipulated. These are significant improvements over previous methods, although there are a few concerns/questions.

1. What is the level of expression of Cx43 in the aCM and can gap junction communication be measured in these cells in culture?
2. To what extent does the use of blebbistatin impair physiological responses to stimuli?
3. Can the cells be passaged, or detached and re-plated for specific experiments?
4. To what extent can 3D culture be achieved with these cells and would that improve overall survival and function?

Specific point-by-point response:

=====

Reviewers' comments:

Reviewer #1 (Remarks to the Author):

1. What are the major claims of the paper?

The authors present a combinatory approach to culturing harvested adult mouse cardiomyocytes (aCMs). Under traditional culture protocols, aCMs rapidly lose their phenotype (e.g., their rod-shaped structure) and undergo cell death. The authors show that with their protocol, aCMs remain viable and retain their phenotype for 7 days - enough time to do genetic manipulation and small molecule screening.

Response: No response required

2. Are they novel and will they be of interest to others in the community and the wider field? If the conclusions are not original, it would be helpful if you could provide relevant references.

Most studies with aCMs are indeed acute studies because of aCM de-differentiate and die in the culture so rapidly. I believe this protocol could be very effective in changing the field.

Response: We appreciate these comments and are grateful for the constructive critiques provided by the reviewer.

3.1 the authors never show the % of viable rod-shaped (aka adult-like) cells under traditional culture conditions. This could be shown in Figure 1 a and c since Fig 1b already show a comparison of Akt activity in aCMs seeded on Laminin or Geltrex

Response: We appreciate these comments and certainly agree that the proposed experiments would provide valuable information. In our submitted revised manuscript, we have added a figure (Now Fig. 2) including this new data. Specifically, in this new figure panel (A) we present data with a full cell count, by day, for each combination of Geltrex/Laminin and BDM/blebbistatin, using the same dishes imaged day by day in the same regions of interest (ROIs). Our results show much greater retention of the adult, rod shaped cells with BDM/blebbistatin compared to conventional methodologies.

3.2 Similarly, the new protocol features a combination of Geltrex and Blebbistatin but it is never shown what are the effect of cell death and de-differentiation of each component.

Response: This is an excellent comment and we agree with the reviewer that this is a very valuable experiment. We have now added panels in a new figure (Now Fig.2) detailing the expanded cell count experiment (See point 3.1 above).

3.3 At various points, the authors reference past published reports on the use of Geltrex and Blebbistatin (or BDM) but never clarify the specific way their approach is different than those previous reports. What is the secret here? The combination? The specific concentrations?

Response: We understand the reviewer's concern and we certainly want to ensure that our specific advancements are clearly stated. In our revised manuscript, we have included critical text (page 2) to explicitly state that the combination of Geltrex and blebbistatin as the key to retaining function in aCMs. A clearer line of evidence is also better established using the inclusion of d7 staining experiment detailed in the next section (See point 3.4 below).

3.4 Another point of confusion is at pag 3, line 53 - where the authors state that previous protocols could not preserve aCMs contractile structure past day 8. Unfortunately, the authors' data show aCMs contractile structure up until day 7, so it's unclear whether they can do better than earlier attempts in this respect. Can this be re-stated? Moreover, this would be much better illustrated with data showing aCMs de-differentiation in laminin-only (traditional) cultures.

Response: We thank the reviewer for highlighting this issue. They are correct in this issue, and in our revised manuscript, we have now added new immunostaining data showing DHPR staining following 7 days culture (with obvious differences between protocols, highlighting the quality of cellular preparations) as well as expanded cell count experiment featuring each of the combinations (new Fig 2b). We believe that this figure clearly articulates that the timecourse and quality of cells is a significant improvement over existing aCM culture methods. We also highlight the limitation in the discussion with appropriate literature (Place et al., 2017, Free Radical Biology and Medicine) and propose moving forward with more physiologically-relevant conditions as a likely outstanding experimental detail to consider (page 9).

3.5 The paragraph continues with a convincing demonstration that the new protocol is better at preserving ultrastructural elements, such as t-tubules, needed for calcium handling. This argument is further discussed convincingly in the following paragraph. Yet, the statements in pag 4, line 79-83 need to be revised. CICR can be triggered by T-Type calcium currents, not just L-type ones (change the word "necessary" with "important"). Moreover, a reference is needed to support the notion that Fluo-4 can be used to track the "delay" calcium fluxes associated with a switch from L- to T-type currents and the loss of t-tubules. In general, this section seems to invoke in an imprecise manner a lot more physiology that's needed to support the simple claim that the new protocol preserves calcium cycling in aCMs better than the old one.

Response: We agree with the reviewer that this text should be improved. We have now made edits in our submitted revised manuscript (page 6) and as the reviewer was correct, we wanted to

state that the methodology better preserves generalized Ca^{2+} transients. Importantly, given the limited electrophysiological workup (and indeed the difficulty in separating co-occurring Ca^{2+} currents), we did not mean to imply that the T-type current indeed becomes more prominent. Given the complexity of all of the various key proteins in regulated EC coupling, we have ensured that we removed any speculation within the text.

3.6 The TFM data is very important, as it is possibly the first time this technique could be used on adult cells. At the same time, one technical question I could not answer in the Methods section is on which day the TFM experiments were performed, and how long the aCMs were kept on TFM substrates before experiments. TFM substrates do not feature GelTrex so it would be important to at least discuss the potential for de-differentiation if aCMs are kept on these substrates for too long.

Response: We appreciate the reviewer's comments and apologize that this was not presented as clearly as we could. The TFM substrates are in fact coated with Geltrex; in this study, it was covalently coupled to the surface of the polyacrylamide using sulfo-SANPAH, although we have also had good results using diluted Geltrex mixed directly into a 2X acrylamide solution prior to crosslinking as normal. TFM was performed at d0 for the substrate stiffness response and Bowditch curve, and at d2 for the PLN experiment to be consistent with the imaging data. We have amended the manuscript to better convey these important details (page 13).

3.7 Also on TFM, the authors compare contractile profiles at 2 and 11 kPa but the interesting comparison would have been between de-differentiated (fetal-like) and preserved aCMs. In fact, 2 kPa is the stiffness of fetal tissues and 11 kPa the stiffness of adult one. That is, do de-differentiated cells contract more strongly on 2 kPa tissues as shown for example by the Parker lab? Alternatively, and as before, I would limit the comparison to the 11 kPa substrates but focus on aCMs cultured with the new protocol and the ones cultured with the old protocol at 1 and 7 days after seeding.

Response: We understand the reviewer's concern. However, we do not observe de-differentiation to a neonatal but functional state with our method in the timescales measured. In the seminal Ackers-Johnson et al. (2016) paper, re-establishment of actinin structure in a stellate (immature) state does not occur until the third week of culture. Additionally, we have not been able to successfully re-establish contraction after removal of BDM using the old method (BDM + laminin as a cell culture substrate coating) after d0, so a paired time course experiment using contraction is not feasible. It is not apparent that Ackers-Johnson et al. were able to do so.

3.8 Fig 3 and 4 showing how the new protocol enables important approaches on physiologically relevant adult CM is spot-on and made even more impressive by the use of novel techniques such as dSTORM.

Response: we are very appreciative of the strong support of the importance of this work.

4. On a more subjective note, do you feel that the paper will influence thinking in the field? Please

feel free to raise any further questions and concerns about the paper.

I like the paper a lot and believe it could be very important for the field, as the types of important experiments described in Fig 3 and Fig 4 are currently possible only on cell lines, neonate primary CM, or stem cell-derived CM - all cell types that do not recapitulate a healthy, adult-like phenotype. At the same time, Fig 1 and 2 only show the good results obtained with the new protocol without providing a direct comparison with the bad outcomes of standard culture conditions or sub-optimal combinations of the two elements they propose. If a stronger basis is provided in Fig 1 and 2, Figures 3 and 4 will have an even bigger impact.

Reviewer: Francesco Pasqualini

Response: Once again, we are grateful for the detailed review of our manuscript and appreciate these comments.

Reviewer #2 (Remarks to the Author):

1. While promising, maintaining aCM cultures for 7 days is a relatively short time span to claim "long-term" culture method. I would suggest that the authors extend the culture duration and characterization to at least 14 days, which will put them in line with drug screening assays and tissue function recapitulation experiments. This will also significantly enhance the impact of this work.

Response: We appreciate these comments and understand the reviewer's concern. We agree with the reviewer that this is an important limitation of the method. Unfortunately, the cells experience high mortality even at d7 (as seen in Fig 1C). To address the reviewer's comments, we identified the mounting free ROS measured using the CellROX assay as a potential mechanism underlying the limitation of functional culture. To this end we attempted supplementing the antioxidant 2-mercaptoethanol that is already present in M199 in addition to our condition panel carried out in response to Reviewer 1's comments. A subtle addition of 2-mercaptoethanol did not influence cell survival, while addition of a moderate dose 2.5 mmol L⁻¹ Tiron (a cell-permeable ROS scavenger) greatly increased cell mortality or detachment (results not shown). For this reason, we suspect that the oxidative state of the cells after prolonged culture is more of an effect, rather than a cause, of cell mortality. As outlined in our response to Reviewer 3 below, we believe that isolated culture, rather than a missing or disproportionate level of some chemical factor, is the chief reason for the divergent function and lowered survival observed. A non-contracting, disconnected state is likely not conducive to continued survival and function of aCMs, as evidenced by the findings in iPSC-CM research showing benefits of connectivity and continual contraction. Nonetheless, we have detailed this explanation in the Discussion (page 9). However, we believe that the functional improvements demonstrated here are still of great interest to researchers in the field. We have added a short discussion on the outer time limit of functionality (page 9).

2. The paper described a method to maintain a difficult cell culture in a more or less functional state, which will help the field where such culture times are necessary. The authors claim that this will help the field recapitulate tissue functions in vitro, but they offer very limited experiments to show this. I would suggest at least a one additional experiment where they use a drug with known effects to compare their culture method to current state of the art culture method.

Response: We appreciate these comments and understand the reviewer's concern. We have now added a panel to now-Figure 3 where we demonstrate increased aCM contractile force in single cells exposed to 5 nM epinephrine (beta-adrenergic agonist) after taking a baseline measurement of the same cell, as assessed by TFM and compared by a paired t-test. This drug treatment represents a physiologically-relevant dose as well as response. In addition, we believe that our other TFM experiments (stiffness sensitivity and Bowditch-like effect) represent important aspects of adult myocardial function that are not appreciably present in other current myocardial models.

3. The statistical analysis, where needed is appropriate and convincing. The work would be possible to replicate given the level of detail provided in the paper and it would be of interest to others in the field. I would recommend to accept with revisions

Response: We appreciate these comments and are grateful for the detailed review of our manuscript.

Reviewer #3 (Remarks to the Author):

These studies establish an improved method for culturing adult cardiomyocytes (aCM) from mice. The investigators introduce the use of Geltrex as an ECM substrate and the use of myosin ATPase inhibitor blebbistatin to reduce aCM contractility on the ECM. They demonstrate that the aCM maintain structure and function over a 7-day period and, during that window, the cells can be genetically and pharmacologically manipulated. These are significant improvements over previous methods, although there are a few concerns/questions.

1. What is the level of expression of Cx43 in the aCM and can gap junction communication be measured in these cells in culture?

Response: We appreciate the reviewer's interest in assessing Cx43 disposition in the cells; this certainly would be a very exciting possibility. In fact, we shared the same interest and indeed had attempted to further investigate Cx43 localization or retention, particularly in conjunction with our previous confluent monolayer/3D tissue experiments detailed more fully below. We have attached to this response some confocal images of Cx43 immunostaining in the isolated myocytes at 1, 24, and 48 h post-isolation. Although intercalated discs are still apparent, it is also evident that there is evidence of Cx43 internalization. These observations are in line with the literature as

dissociating the cells is known to lead to protein degradation and breakdown of the ICD (Thévenin et al. [2013] Physiology 28(2):93-116; Mazet et al. [1985] Circ Res 56(2):195-204; Severs et al. [1989] 65(1):22-42).

2. To what extent does the use of blebbistatin impair physiological responses to stimuli?

Response: As blebbistatin inhibits contraction at the point of myosin (myosin ATPase activity) (Kovacs et al. [2004] J Biol Chem 279:35557-63), our contention is that it preserves responses to purely electrophysiological or chemical stimuli before that point (i.e. Ca^{2+} handling is preserved). Continued contractility is well-known to be important for CM homeostasis through a range of mechanisms. For this reason, we believe that the lack of contraction, cell-cell impulse propagation, syncytial passage of intracellular molecules, and associated cyclic mechanical signaling (e.g. through integrins and actin-associated signaling pathways) would result in a progressive response that deviates from in vivo or

freshly ex vivo physiology. We have clarified this in the discussion in our submitted revised manuscript (page 9).

3. Can the cells be passaged, or detached and re-plated for specific experiments?

Response: Propagating or passaging would be a novel aspect of this work. However, we have not observed proliferation of the cells in culture as adult cardiomyocytes are a terminally differentiated cell population. This is congruent with maintained maturity and rod-shaped structure, as mitotic cytokinesis would necessarily result in loss of sarcomeric organization. As for replating, any attempts to detach cells typically results in substantial mortality even at d0. This could likely be attributed to hypercontraction as discussed previously pertaining to the dissociation from the heart during isolation, as well as continued handling and shear forces.

4. To what extent can 3D culture be achieved with these cells and would that improve overall survival and function?

Response: We appreciate these comments and agree with the reviewer that this would be the most likely way to further enhance aCM survival and function. To this end, we have attempted both 3D and confluent monolayer culture with aCMs but have had little success in either. For 3D culture, cells do not align nor contract (by both continual force and active remodeling) a gel, and so are not internally “confluent” such that they mimic myocardium; in current iPSC-CM 3D culture such as the Biowire II (Zhao et al. 2019) system, contraction of the gel into a cohesive tissue is mostly performed by fibroblasts in co-culture. In monolayer culture, when seeding cells densely, we find that the aCMs tend to attach to each other and not the cell culture substrate. This attachment is not regulated in any way, such that the CMs are unaligned. The use of plating forms, such as microtrenches or gratings, aligns cells to an extent but does not ameliorate the issue of cell-cell interference during plating. Given our lack of progress in this area, we could not find significant value in continuing these experiments. A summary of our findings has been included in the discussion (page 9).

END OF COMMENTS

=====

REVIEWERS' COMMENTS:

Reviewer #1 (Remarks to the Author):

I thank the authors for the additional experiments and discussion points - the addition of the new figure 2 really helps clarify the manuscript and better articulate the importance of the developed protocol.

A couple of minor suggestions:

1. Split the rod- and round-shape curves in Fig 2a so that the traces are more clearly visible
2. Add statistical information. It should be noted whether the results are statistically significant. Even if not all results are statistically significant, the authors can acknowledge the trend and hint to typical issues with sample size and the innate variability in primary preparations

Reviewer #3 (Remarks to the Author):

This manuscript is sufficiently revised and will be useful for the field.

Specific point-by-point response:

=====

Reviewers' comments:

Reviewer #1 (Remarks to the Author):

I thank the authors for the additional experiments and discussion points - the addition of the new figure 2 really helps clarify the manuscript and better articulate the importance of the developed protocol.

A couple of minor suggestions:

1. Split the rod- and round-shape curves in Fig 2a so that the traces are more clearly visible

Response: We have revised the figure as suggested.

2. Add statistical information. It should be noted whether the results are statistically significant. Even if not all results are statistically significant, the authors can acknowledge the trend and hint to typical issues with sample size and the innate variability in primary preparations

Response: We have added factorwise comparisons post 2-way RM ANOVA to the figure.

Reviewer #3 (Remarks to the Author):

This manuscript is sufficiently revised and will be useful for the field.

END OF COMMENTS

=====